# Glycosaminoglycans as Biomarkers for Mucopolysaccharidoses and Other Disorders

**DOI:** 10.3390/diagnostics11091563

**Published:** 2021-08-28

**Authors:** Paige C. Amendum, Shaukat Khan, Seiji Yamaguchi, Hironori Kobayashi, Yasuhiko Ago, Yasuyuki Suzuki, Betul Celik, Estera Rintz, Jobayer Hossain, Wendi Xiao, Shunji Tomatsu

**Affiliations:** 1Department of Biological Sciences, University of Delaware, Newark, DE 19716, USA; pamendum@udel.edu (P.C.A.); Betul.Celik@nemours.org (B.C.); 2Department of Biomedical Research, Nemours/Alfred I. duPont Hospital for Children, Wilmington, DE 19803, USA; Shaukat.Khan@nemours.org (S.K.); Estera.Rintz@nemours.org (E.R.); MD.Hossain@nemours.org (J.H.);; 3Department of Pediatrics, Shimane University, Izumo 693-8501, Japan; seijiyam@med.shimane-u.ac.jp (S.Y.); bakki@med.shimane-u.ac.jp (H.K.); 4Department of Pediatrics, Graduate School of Medicine, Gifu University, Gifu 501-1193, Japan; Yasuhiko.Ago@nemours.org; 5Medical Education Development Center, Graduate School of Medicine, Gifu University, Gifu 501-1193, Japan; ysuz@gifu-u.ac.jp; 6Department of Pediatrics, Thomas Jefferson University, Philadelphia, PA 19107, USA

**Keywords:** glycosaminoglycan, encephalopathy, serum, LC-MS/MS, mucopolysaccharidoses

## Abstract

Glycosaminoglycans (GAGs) are present in proteoglycans, which play critical physiological roles in various tissues. They are known to be elevated in mucopolysaccharidoses (MPS), a group of rare inherited metabolic diseases in which the lysosomal enzyme required to break down one or more GAG is deficient. In a previous study, we found elevation of GAGs in a subset of patients without MPS. In the current study, we aim to investigate serum GAG levels in patients with conditions beyond MPS. In our investigated samples, the largest group of patients had a clinical diagnosis of viral or non-viral encephalopathy. Clinical diagnoses and conditions also included epilepsy, fatty acid metabolism disorders, respiratory and renal disorders, liver disorders, hypoglycemia, developmental disorders, hyperCKemia, myopathy, acidosis, and vomiting disorders. While there was no conclusive evidence across all ages for any disease, serum GAG levels were elevated in patients with encephalopathy and some patients with other conditions. These preliminary findings suggest that serum GAGs are potential biomarkers in MPS and other disorders. In conclusion, we propose that GAGs elevated in blood can be used as biomarkers in the diagnosis and prognosis of various diseases in childhood; however, further designed experiments with larger sample sizes are required.

## 1. Introduction

In the human body, carbohydrates exist as GAGs: sulfated polysaccharide chains. These GAG chains attach to core proteins, forming proteoglycans (PGs), which have various functions, including cell signaling, stimulating growth and development, and extracellular matrix (ECM) hydration. The core proteins of proteoglycans can be transmembrane; therefore, GAGs can be a part of the ECM or part of the glycocalyx. GAGs include chondroitin sulfate (CS), dermatan sulfate (DS), heparan sulfate (HS), keratan sulfate (KS), and hyaluronic acid (HA). HA differs from the other GAGs, as it is neither sulfated nor linked to a core protein [1].

Each GAG chain is found in different cells and tissues and has various roles in the central nervous system (CNS), visceral organs, and connective tissues. Chondroitin sulfate proteoglycans (CSPGs), which commonly contain both CS and DS, are the most abundant proteoglycan in the CNS. CSPGs typically act as barrier molecules, directing axon growth and synapse formation. The PGs of the lectican family, which contain mainly CS as well as KS [2], are the main constituents of the brain ECM. These PGs bind with HA and link proteins in the brain ECM with neurons [3]. Like CSPGs, KS proteoglycans (KSPGs) in the CNS are also mainly involved in neuronal outgrowth and synapse organization. Additionally, KSPGs also play a role in neurotransmission and nerve regeneration [2]. Both CS and KS are known to play a role in glial scarring and regeneration following brain injury. CS and DS have also been shown to bind to morphogens, making them essential in CNS development and mediating cell proliferation [4]. The main function of HA in the CNS is its structural role in forming the brain ECM, but it has also been shown to bind to growth factors and cytokines. Additionally, low molecular weight HA is involved with inflammation after CNS injury [4]. Finally, HS proteoglycans (HSPGs) constitute a significant component of the vascular basement membrane in brain [5]. They bind with signaling molecules, preventing their degradation and creating storage pools. HSPGs also form ternary complexes with signaling molecules and their receptors to promote signaling.

Mucopolysaccharidoses (MPS) are a group of inherited metabolic disorders in which patients have a deficiency of a lysosomal enzyme required to degrade one or more GAG, leading to an accumulation of GAGs in the lysosomes. This accumulation interrupts normal cell physiology, resulting in a complex syndrome with symptoms including skeletal dysplasia, organ dysfunction, developmental delay, cognitive impairment, hearing loss, and joint rigidity or hypermobility. Currently, enzyme replacement therapy and hematopoietic stem cell transplantation are available clinically for MPS. Both treatments provide a better prognosis if patients are treated at an early age. DS, HS, and KS are commonly used as biomarkers for high-risk or newborn screening of MPS [6,7].

Previous studies have shown that some PGs are elevated or altered in various specimens (urine, blood, cerebrospinal fluid, tissues) in some diseases [8,9]. For example, the DSPG endocan is elevated in patients with stable chronic obstructive pulmonary disease (COPD) (*n* = 47) [8], and syndecan-4, an HSPG, is elevated in response to bacterial pneumonia (*n* = 30) [9]. Further studies have demonstrated the elevation of specific GAGs in some diseases or conditions, mainly in adulthood. For instance, GAGs constitute a large part of the endothelial glycocalyx in the vascular lumen, which is perturbed in illnesses with systematic inflammation, such as respiratory failure or septic shock, leading to an increase in highly sulfated HS fragments in the blood (*n* = 17) [10]. The endothelial glycocalyx has also been implicated in post-cardiac arrest syndrome in adults, indicating that cardiac arrest or resuscitations can shed the glycocalyx components syndecan-1, HS, and HA into blood circulation. Additionally, patients who survive cardiac arrest have lower HS and syndecan-1 levels than deceased patients, indicating that the extent of glycocalyx perturbation and corresponding levels of GAGs in blood could indicate prognosis in these adult patients (*n* = 25) [11]. However, there has not been a study concerning the measurement of each GAG in blood specimens covering a wide range of common diseases and inherited metabolic disorders, especially in childhood.

In previous studies investigating GAG levels in blood or dried blood spots (DBS) of MPS patients, we found elevated GAG levels in a subset of control individuals as well as in patients confirmed to have MPS [7,12]. It was determined that these control samples did not come from MPS patients with clinical findings and enzyme activity assays. Therefore, it is essential to discover which diseases and conditions cause the elevation of GAGs, determine which GAGs are elevated in each, and explore whether the measurement of GAGs is a valuable tool for disease prognosis and monitoring therapeutic effects.

In the present study, the data of patients with conditions including MPS and other diseases have been grouped according to clinical diagnosis. The clinical diagnosis groups include respiratory and renal disorders, fatty acid metabolism disorders, viral infections, vomiting disorders, liver disorders, epilepsy, hypoglycemia, myopathy, developmental disorders, hyperCKemia, heart disease, acidosis, and encephalopathy. We aim to investigate GAG expression in the blood of pediatric patients with various conditions and compare the levels of circulating GAGs to the levels of patients with MPS and age-matched controls. In summary, this is a novel study involving patients with various diseases to determine if GAGs can be used as biomarkers.

## 2. Materials and Methods

### 2.1. Subjects

For patients and control subjects, we measured two types of HS—*O*-sulfated HS (∆DiHS-0S) and *N*-sulfated HS (∆DiHS-NS), DS (∆Di-4S), two types of KS—mono-sulfated KS and di-sulfated KS, and the ratio of di-sulfated KS to total KS. These are the GAGs that are measured to screen for MPS [6,7,12].

This was a retrospective study using clinical data and serum samples from Shimane University. Both male and female patients were included. All patients were Japanese; therefore, although we expect that we would see similar results across other populations, we cannot draw conclusions about similarities in people of different ethnic backgrounds based on the current data. Serum samples were obtained with informed consent from 276 patients with various clinical conditions and diagnoses (Appendix A). Patient ages ranged from zero to sixty-two years; however, only 14 patients (5.1%) were over the age of fifteen, so that this was primarily a pediatric study. Among the patients with various clinical conditions, 140 patients (51%) were 0–2.9 years old, 35 patients (13%) were 3–4.9 years old, 53 patients (19%) were 5–9.9 years old, 30 patients (11%) were 10–14.9 years old, 7 patients (2.5%) were 15–19.9 years old, and 11 patients (4.0%) were 20 years old or older. Clinical diagnosis and enzyme activity assays confirmed that patients did not have MPS. Using the clinical diagnoses provided by Shimane University, patient data were sorted into groups according to diagnosis. Clinical diagnoses for respiratory or renal disorders included pneumonia, asthma, bronchitis, chronic obstructive pulmonary disease (COPD), and rhabdomyolysis. Fatty acid metabolism disorders diagnoses included carnitine deficiency, Reye’s syndrome, carnitine palmitoyltranferase 2 (CPT2) deficiency, medium-chain acyl-CoA dehydrogenase (MCAD) deficiency, and very-long-chain acyl-CoA dehydrogenase (VLCAD) deficiency. Viral infections included rotavirus, hand-foot-mouth disease, and influenza. Vomiting disorders were all cyclic vomiting syndrome. Clinical diagnoses for liver disorders included hyperbilirubinemia and liver dysfunction. Clinical diagnoses for epilepsy included West syndrome, tonic-clonic seizures, and febrile seizures. Heart conditions included hypertrophic cardiomyopathy, abnormal ECG, mitral regurgitation (MR), myocarditis, and ventricular tachycardia. Clinical diagnoses for acidosis included glutaric acidemia II (GAII) and methylmalonic acidemia. Viral encephalopathy viruses included respiratory syncytial virus (RSV), influenza A, influenza B, rotavirus, human herpesvirus 6 (HHV-6), and norovirus. Non-viral encephalopathy diagnoses included megalencephaly, hypoglycemia encephalopathy, epileptic encephalopathy, hypoxic-ischemic encephalopathy, Leigh syndrome, periventricular leukomalacia, ifosfamide-induced encephalopathy, acute focal bacterial nephritis (AFBN) encephalopathy, and leukoencephalopathy. Twenty-two patients were clinically diagnosed with respiratory or renal conditions, 21 patients were diagnosed with some sort of fatty acid metabolism disorder, 7 patients were diagnosed with viral infections without encephalopathy or other symptoms, 13 patients were diagnosed with vomiting disorders, 18 patients were diagnosed with liver disorders, 33 patients were diagnosed with epilepsy, 22 patients were diagnosed with hypoglycemia, 12 patients were diagnosed with myopathy, 14 patients were diagnosed with developmental disorders, 12 patients were diagnosed with hyperCKemia, 15 patients were diagnosed with a heart condition, 16 patients were diagnosed with acidosis, 51 patients were diagnosed with viral encephalopathy, and 69 patients were diagnosed with non-viral encephalopathy. The total number of patients in these groups adds up to more than 276 patients because some patients had overlapping conditions and were thus used in more than one group.

We compared the GAG levels of patients with various clinically diagnosed conditions to the levels of patients with MPS (Appendix A). Blood samples from patients with MPS were collected with informed consent at Gifu University. MPS patients were from various ethnic backgrounds and included both male and female patients. The breakdown of MPS type and age is shown below (Table 1).

Control values were obtained from data collected for a previous study [12]. The control values were obtained with informed consent from patients at Shimane University (Appendix A). The breakdown of age for the control values is shown below (Table 2).

We also collected 198 dried blood spot samples from control newborns, including one MPS II patient, in a double-blind manner. Procedures were approved by IRBs at Nemours/AIDHC (approval number: 281498-21).

### 2.2. Enzymes and Standards

Enzymes and stock solutions used to make standards were obtained from Seikagaku Corporation (Tokyo, Japan). Heparitinase, chondroitinase B, and keratanase II were used to digest the polysaccharide GAG chains into their respective disaccharides: 2-deoxy-2-sulfamino-4-(4-deoxy-a-l-threo-hex-4-enopyranosyluronic acid)-d-glucose (∆DiHS-NS), 2-acetamido-2-deoxy-4-O-(4-deoxy-a-l-threo-hex-4-enopyranosyluronic acid)-d-glucose (∆DiHS-0S), 2-acetamido-2-deoxy-4-O-(4-deoxy-a-l-threo-hex-4-enopyranosyluronic acid)-4-O-sulfo-d-glucose (∆Di-4S; DS), mono-sulfated KS (Galß1-4GlcNAc(6S)), and di-sulfated KS (Gal(6S)ß 1-4GlcNAc(6S)). Stock solutions of the above disaccharides were used to make standard solutions by serial dilution consisting of 1000 ng/mL, 500 ng/mL, 250 ng/mL, 125 ng/mL, 62.5 ng/mL, 31.25 ng/mL, 15.625 ng/mL, and 7.8125 ng/mL of ∆DiHS-NS, ∆DiHS-0S, and ∆Di-4S, and 10,000 ng/mL, 5000 ng/mL, 2500 ng/mL, 1250 ng/mL, 625 ng/mL, 312.5 ng/mL, 156.25 ng/mL, and 78.125 ng/mL mono-sulfated KS and di-sulfated KS. Chondrosine was used as an internal standard.

### 2.3. Sample Preparations

In AcroPrep^TM^ Advance 96-Well Filter Plates with Ultrafiltration Omega 10K membrane filters (Pall Corporation, Port Washington, NY, USA), in order, the following was added: 10 microliters of sample or standard; 90 microliters of 0.5 M Tris buffer pH 7.0; 40 microliters of a cocktail consisting of heparitinase (0.5 mU/sample), chondroitinase B and keratanase (both 1 mU/sample), and internal standard (5 μg/mL); 60 microliters of 0.5 M Tris buffer pH 7.0. The filter plate was incubated overnight on a 96-well receiver plate at 37 °C to digest the polysaccharides. The filter plate was then placed on a new receiver plate and centrifuged for 15 min at 2500 rpm to filter the digested disaccharides. The processed samples were injected and measured using liquid chromatography-tandem mass spectrometry (LC-MS/MS).

### 2.4. LC-MS/MS

The chromatographic system used has been described in earlier studies [12,13,14,15,16,17]. The mobile phases were 100 mM ammonia (A) and 100% acetonitrile (B). The initial composition of 100% A was held for 1 min, linearly modified to 30% B at 4 min, maintained at 30% B at 5.5 min, returned to 0% B at 6 min, and maintained at 0% B until 10 min. The flow rate was 0.7 milliliter per minute. DS was measured as Di-0S due to digestion of Di-4S to Di-0S by a 4S-sulfatase present in the chondroitinase B. The concentration of each disaccharide was calculated by QQQ Quantitative Analysis software.

### 2.5. Statistical Analysis

This is an observational study to investigate serum GAG levels in patients with MPS and encephalopathy (viral or non-viral) and compare with the levels in control subjects. Patient data were grouped according to diagnosis or condition. Since GAG levels are also influenced by age, patient data were then divided into the following age groups: x < 3 years, 3 ≤ x < 5 years, 5 ≤ x < 10 years, 10 ≤ x < 15 years, and over 15 years of age. Examination of box plots exhibited a similar shape in the distribution of GAGs, except a few sparse outliers, across diagnostic groups for each age category. A non-parametric Kruskal–Wallis one-way analysis of variance (ANOVA) was performed to compare median GAG levels between ten groups for each GAG at each age group, including the control group, viral encephalopathy patients, non-viral encephalopathy patients, MPS I patients, MPS II patients, MPS IIIA patients, MPS IIIB patients, MPS IVA patients, MPS IVB patients, and MPS VII patients.

Additionally, a Dunn’s post-hoc test was used to determine statistical significance between the control and encephalopathy as well as between the control and MPS groups. All tests were two-tailed at the overall level of significance of 0.05. The statistical software packages R, version 3.5.2, and SPSS, version 27, were used for data analyses (Armonk, NY, USA). Despite well-known limitations, data from observational studies have become an increasingly important source of evidence, and thus, these results have relevance [18].

## 3. Results

### 3.1. Conditions with Elevated GAG Levels

We detected high levels of DiHS-0S, DiHS-NS, Di-4S, di-sulfated KS, and di-sulfated KS/total KS in some patients with viral encephalopathy and some patients with non-viral encephalopathy. A patient with heart disease had high levels of DiHS-0S, DiHS-NS, and Di-4S. A patient with a viral infection had high levels of DiHS-0S and DiHS-NS. Some patients with epilepsy had high levels of DiHS-0S, DiHS-NS, di-sulfated KS, and di-sulfated KS/total KS. A patient with a developmental disorder had high levels of DiHS-NS. Some patients with hypoglycemia had high levels of Di-4S, mono-sulfated KS, di-sulfated KS, and di-sulfated KS/total KS. A patient with fatty acid metabolism disorders had high levels of di-sulfated KS. Some patients with acidosis had high levels of Di-4S, di-sulfated KS, and di-sulfated KS/total KS. However, the sample size in most groups was too small to run statistical analyses. Only the encephalopathy groups had large enough sample sizes to conduct further statistical analysis.

### 3.2. Encephalopathy GAG Levels

We compared the median, minimum, and maximum values for the control group, viral encephalopathy group, non-viral encephalopathy group, and various MPS types (Table 3).

There is some variation in median GAG levels across age groups. However, there are significant differences in all GAGs measured between the control, viral encephalopathy, non-viral encephalopathy, MPS I, MPS II, MPS IIIA, MPS IIIB, MPS IVA, and MPS IVB. Of particular note is the 0–2.9 age group, which only contained the control and two encephalopathy groups. In this comparison, there is a significant difference in DiHS-0S, DiHS-NS, Di-4S, di-sulfated KS, and di-sulfated KS/total KS between the control group, viral encephalopathy group, and non-viral encephalopathy group (Table 3). Using Dunn post-hoc testing for pairwise comparisons in the control group, viral encephalopathy had significantly higher DiHS-0S, DiHS-NS, Di-4S, and di-sulfated KS medians for the 0–2.9 age group and the 10–14.9 age group and a significantly higher ratio of di-sulfated KS to total KS for the 0–2.9 age group, the 5–9.9 age group, and the 10–14.9 age group. Non-viral encephalopathy had significantly higher DiHS-0S, Di-4S, and di-sulfated KS for the 0–2.9 age group, significantly higher DiHS-NS for the 0–2.9 age group and the 5–9.9 age group, and a significantly higher ratio of di-sulfated KS to total KS for the 0–2.9 age group, 3–4.9 age group, and the 5–9.9 age group. None of the age groups had a statistically significant difference in average mono-sulfated KS values for either type of encephalopathy.

### 3.3. GAG Levels of Various Conditions

The y-axes for Figure 1, Figure 2, Figure 3, Figure 4, Figure 5 and Figure 6 have been log-transformed due to extreme outliers. Without the transformations, outliers made it difficult to see the distributions of GAG measurements. HS and DS decrease slightly with age (Figure 1, Figure 2 and Figure 3), while mono-sulfated and di-sulfated KS drop rapidly (Figure 4 and Figure 5). The ratio of di-sulfated KS to total KS is the only value that increases with age (Figure 6).

Some patients with viral encephalopathy, non-viral encephalopathy, heart disorders, and viruses had serum ∆DiHS-0S well above control values. These patients had higher ∆DiHS-0S levels than MPS patients at a similar age (Figure 1). This is unexpected, as HS accumulates in patients with MPS I, MPS II, MPS III and MPS VII as a primary storage material [19].

Some patients with viral encephalopathy, non-viral encephalopathy, heart disorders, viruses, epilepsy, and developmental disorders had serum ∆DiHS-NS well above control values. Furthermore, some patients with viruses, viral encephalopathy, non-viral encephalopathy, and heart disorders had higher ∆DiHS-NS levels than MPS patients at a similar age (Figure 2). Typically, HS accumulates in patients with MPS I, MPS II, MPS III and MPS VII as a primary storage material [19].

Some patients with viral encephalopathy, non-viral encephalopathy, heart disorders, hypoglycemia, epilepsy, and acidosis had serum Di-4S well above control values. However, none of these diagnosis groups had Di-4S levels higher than MPS II (Figure 3).

Some patients with viral encephalopathy and hypoglycemia had higher mono-sulfated KS values than controls. Most MPS II and some MPS IVA patients had mono-sulfated KS values higher than the control values, the viral encephalopathy values, and the hypoglycemia values (Figure 4). KS is known to accumulate in patients with MPS IV as a primary storage material [19].

Some patients with viral encephalopathy, non-viral encephalopathy, fatty acid metabolism disorders, hypoglycemia, epilepsy, and acidosis had serum di-sulfated KS well above control values. Furthermore, some patients with viral encephalopathy, acidosis, and epilepsy had higher di-sulfated KS values than patients with MPS at a similar age (Figure 5). This is unexpected since KS accumulates in patients with MPS IV [19].

Some patients with viral encephalopathy, non-viral encephalopathy, hypoglycemia, epilepsy, and acidosis had a ratio of serum di-sulfated KS to total serum KS well above the control values. Furthermore, most patients also had higher ratio values than patients with MPS at a similar age (Figure 6).

### 3.4. GAG Levels of Newborns

Out of 198 DBS samples, two samples provided a significant elevation of specific GAGs. The sample with MPS II showed that the concentration levels of Di-0S, HS-0S, HS-NS, mono-sulfated KS, and di-sulfated KS were 30.9 ng/mL, 141.6 ng/mL, 26.04 ng/mL, 146.0 ng/mL, and 35.6 ng/mL, respectively [7]. HS-0S and HS-NS levels were above the established cutoff values of 90 ng/mL and 23 ng/mL, respectively [6]. Another sample was derived from an extremely premature infant with extremely low birth weight (birth weight; 582 g at 24 gestational weeks, female). The concentration levels of Di-0S, HS-0S, HS-NS, mono-sulfated KS, and di-sulfated KS were 137.3 ng/mL, 104.6 ng/mL, 12.6 ng/mL, 187.6 ng/mL, and 39.2 ng/mL, respectively. Di-0S and HS-0S levels were above the established cutoff values of 88 ng/mL and 90 ng/mL, respectively.

## 4. Discussion

This is, to the best of our knowledge, the first study to measure blood GAG levels in patients with a wide range of pediatric diseases and conditions. We have demonstrated the elevation of serum GAGs in conditions beyond MPS. GAG elevation was seen in a group of patients with encephalopathy, patients with other childhood disorders, and an extremely premature infant.

Note that the control GAG values were represented with a linear model (Figure 1, Figure 2, Figure 3, Figure 4, Figure 5 and Figure 6). The y-axes were log-transformed, so any linear relationship on the graphs would have implied an exponential relationship. The coefficients of determination were low for all models, but this is not an issue, as we were not searching for an exponential relationship since GAGs do not increase exponentially with age; rather, they display the average GAG level across ages.

Any statistical significance of the non-parametric Kruskal−Wallis ANOVA test indicates that the variation mostly explains variations in GAG levels due to groups rather than residuals. We saw statistical significances for the non-parametric Kruskal−Wallis ANOVA test for all GAGs tested in at least five age groups; however, these comparisons also included MPS patients, so it is expected that there would be a statistical significant difference, and we cannot conclude that the encephalopathy group caused this significance. The 0–2.9 age group only contained the control and two encephalopathy groups. In this comparison, there is still a significant difference in DiHS-0S, DiHS-NS, Di-4S, di-sulfated KS, and di-sulfated KS/total KS when only comparing the control group, viral encephalopathy group, and non-viral encephalopathy group (Table 3).

PGs and GAGs are altered in some diseases in quality and quantity [8,9,10,20,21,22]. In particular, conditions in which the endothelial glycocalyx is damaged result in the elevation of plasma and serum GAGs. Serum HS has been found to be elevated in respiratory failure due to indirect lung injury, while serum HA was elevated following direct lung injury in adulthood (age range: 32–68 years); additionally, the persistence of elevated HS in the blood days after injury could indicate impaired GAG clearance as well as glycocalyx degradation [10]. Adult patients (average age: 67.1 ± 3.1 years) with bacterial pneumonia were found to have elevated serum syndecan-4 levels (an HSPG). Additionally, serum syndecan-4 levels had a negative correlation with severity, and syndecan-4 knockout mice had higher mortality rates than control mice when infected, suggesting that syndecan-4 plays a protective role in bacterial pneumonia [9]. Serum endocan, a DSPG expressed in endothelial cells of the lungs and kidneys, is elevated in adult patients (age range: 40–75 years) with COPD and could serve as a severity marker for acute inflammatory lung diseases with endothelial involvement [8]. Serum HS and HA were elevated in adult patients (age range; 45–86 years) with septic shock (*n* = 24) [20]. Therefore, we hypothesized that GAG levels would be elevated in the respiratory and renal conditions group even in childhood. Surprisingly, we did not see any patients with high serum GAGs in this group. However, we did see high serum GAGs in patients with viral encephalopathy, non-viral encephalopathy, a heart disorder (heart disease), viral infections, epilepsy, a developmental disorder, hypoglycemia, fatty acid metabolism disorder, and acidosis. However, only the encephalopathy group had a sample size large enough to run statistical analyses; therefore, further investigation is needed for conclusive evidence that all other conditions are correlated with an upregulation in the serum GAG level.

A patient with a viral infection had high levels of DiHS-0S and DiHS-NS (Figure 1 and Figure 2). The potential mechanism of GAG elevation caused by viral infections remains unknown; however, GAGs have been shown to facilitate the binding and entry of viruses into cells during infection. HS has been established as the initial point of interaction for hepatitis B and herpes [23,24]. These interactions between HS and viruses could play a role in elevating serum HS in patients with viral infections.

Some patients with epilepsy had high levels of DiHS-0S, DiHS-NS, di-sulfated KS, and di-sulfated KS/total KS (Figure 1, Figure 2, Figure 5 and Figure 6). MPS patients often experience epileptic seizures because of the buildup of GAGs in the brain [21]. However, GAGs seem to play a different role in patients with epilepsy: CSPGs are altered. In the adult CNS, PGs and proteins condense around strategic locations—the best-studied location being perineuronal nets (PNNs), lattice-like structures that surround cells, most often parvalbumin-expressing inhibitory neurons (PV-cells) [3]. Neuronal remodeling is seen in patients with epilepsy, and the number of PNNs is decreased. This disruption causes dysregulation in the excitation-inhibition balance in the brain. Seizures also may lead to the degradation of aggrecan, a CS- and KS-rich proteoglycan, by matrix metalloproteinases (MMPs) [1]. Additionally, a decrease in the CS and KS proteoglycan phosphacan corresponding with a decrease in PNNs and an increase in the CSPG neurocan was observed in a rat model with induced seizures [22]. The effect of epilepsy on serum GAG levels has not been widely studied. However, based on previous studies, we speculate that neuronal remodeling and successive degradation of PGs could increase serum GAGs levels, especially CS and KS.

The encephalopathy groups had larger sample sizes, and, therefore, statistical analyses were run between viral encephalopathy, non-viral encephalopathy, and a control group. Serum GAG levels are related to encephalopathy (Table 3); however, there was a lack of conclusive evidence across age groups, likely because some age groups included relatively small numbers of patients and/or control samples. Larger samples are required across age groups. Notably, some patients with encephalopathy had a more significant elevation than MPS patients.

Encephalopathy is any disease or injury that affects the structure or function of the brain. Many events can cause encephalopathy, including infection, tumor, and stroke. While there are no previous studies concerning serum GAG levels in encephalopathy, there are some studies that deal with these underlying causes. Patients had an increase in serum CS, HS, and KS seven days after an ischemic stroke, and these GAG levels returned to baseline ninety days later. These results also showed sheddase activity and found that GAG chains shed before proteoglycans (*n* = 9–14) [25]. This finding suggests that patients with encephalopathy caused by stroke may have elevated serum CS, HS and KS levels as well. Additionally, upregulation of heparanase in vascular cells and astrocytes was found in a mouse model following stroke. The upregulation of heparanase, along with the degradation of HSPGs, may correspond to angiogenesis and tissue repair in the brain following stroke [26].

Previous studies explored elevated GAG levels and/or related PGs in astrocytes, neurons, and endothelial cells after brain injury, often in mouse or rat models [27,28,29]. McKeon et al. discovered elevated levels of a CS/KS-PG and cytotactin/tenascin in astrocytes corresponded with glial scarring in a rat model after cerebral cortex injury [27]. This is unsurprising, as the primary roles of KSPGs and CSPGs in the CNS are directing neuron growth and synapse organization [2,3]. Additionally, Kato et al. found upregulated KS expression in the hippocampus of patients with astrocytic tumors, suggesting that KS may relate to malignancy in tumors [28]. Leadbeater et al. found an upregulation of HSPGs in astrocytes, neurons, and endothelial cells after cerebral cortex injury in a rat model associated with an upregulation in fibroblast growth factor 2 and fibroblast growth factor receptor 1 [29]. This association suggests that HSPGs such as syndecan-2, glypican-1, perlecan, and syndecan-3 are involved with angiogenesis and tissue regeneration after brain injury. The remodeling of CSPGs, HSPGs, and KSPGs in the brain following injury and disease could be related to the elevation of serum HS and KS in patients with encephalopathy seen in this study. However, the mechanism of GAG elevation associated with encephalopathy remains unknown, so an elevation of circulating GAG fragments may cause encephalopathy. For example, sepsis causes the shedding of the endothelial glycocalyx, and circulating HS has contributed to cognitive impairment in sepsis patients (*n* = 17) [30]. Hippensteel et al. concluded that circulating HS may be used as a biomarker to identify adult septic patients (age range: 41–64.5 years) at high risk for cognitive impairment. 

We have also demonstrated that the extremely premature infant provides a significant DS and HS elevation in the DBS sample. The DS level was much higher than that in a severe form of MPS II. The indexed case had normal development after birth at 3 years of age without any brain damage. It is of great interest to understand whether premature brain and body at the developing stage need more specific GAGs with a high level of GAGs in blood or not. Further investigation with more DBS samples at newborns is under development.

There are several limitations for the current study. First, it was a retrospective study. Although the encephalopathy group included the analysis of the largest number of blood samples (*n* = 120 patients; viral, 51, non-viral, 69) compared to previous studies of other disorders (*n* = 8–47 patients) [8,9,10,11,12,25,28,30,31], most sample groups were small, affecting the statistical analysis. Further experiments with larger sample sizes are necessary to confirm these results. Additionally, encephalopathy is often caused by an underlying disease or condition, so it is unclear whether the underlying cause affects GAG levels or encephalopathy itself does. Furthermore, some patients had multiple diagnoses, and were thus included in both groups. In these cases it is not clear which diagnosis would cause GAG elevation. This relates to another limitation of the study. Due to this being a retrospective study, it is impossible to distinguish whether any elevated GAGs preceded the condition, making GAGs an excellent candidate for biomarkers, or if elevated GAGs are symptoms following the onset of a condition. To truly make this distinction, further studies focusing on a single disease or condition would need to investigate GAG levels in a mouse model throughout disease progression. However, if elevated GAGs are seen in mice before the onset of symptoms, as in MPS, then GAGs may be a good candidate for biomarkers.

Based on each GAG’s function in the CNS, we hypothesize that DS and KS elevation may be related to scarring in the brain post-injury, while HS elevation may be related to angiogenesis and tissue repair mechanisms. While it does have limitations, this study provides insight into the level of GAGs in blood associated with various diseases, especially in pediatric patients following brain-associated disease.

We have demonstrated that some patients have an elevation of one or more GAG, especially patients with encephalopathy. This is a significant finding, as GAGs thus have potential to be used as a biomarker for encephalopathy. While this result shows promise, this is still a preliminary investigation of multiple conditions. Further studies into disease progression and severity focusing on a single condition would be necessary to establish GAGs as biomarkers for other diseases.

## 5. Conclusions

Patients with MPS are known to have high levels of serum GAGs due to the accumulation of GAGs in the lysosome. Patients with other conditions also have an elevation of GAGs. Viral and non-viral encephalopathy are associated with elevated GAG levels. This can make GAGs useful biomarkers for various conditions beyond MPS.

The mechanism of GAG elevation in patients with encephalopathy remains unknown. Further studies into the disease progression of encephalopathy are necessary to confirm that GAG elevation precedes the onset of major symptoms of encephalopathy. This is a retrospective study limited by the sample sizes of disease groups, so the results give a preliminary look into various conditions associated with GAG elevation. Further studies into the effect of disease severity and the relationship of GAGs with different diseases and conditions are required before GAGs can be established as biomarkers for the diagnosis and prognosis of the disease. These findings create a foundation for further exploration of elevated serum GAGs in diseases beyond MPS.

## Figures and Tables

**Figure 1 diagnostics-11-01563-f001:**
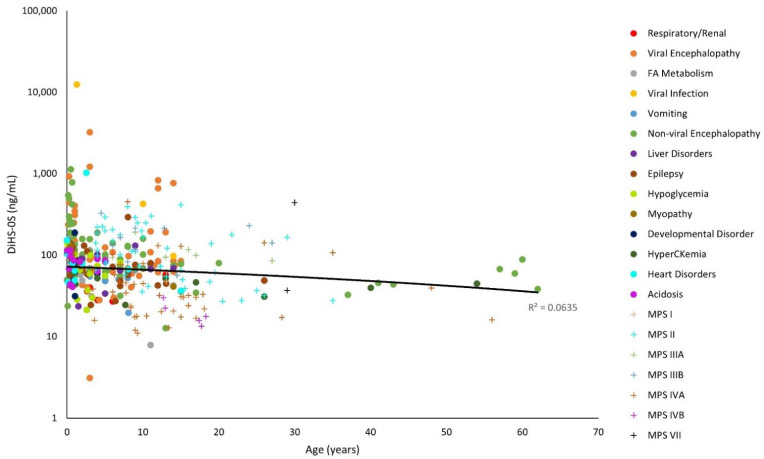
*O*-sulfated heparan sulfate (∆DiHS-0S) levels with age on a logarithmic scale. The linear trendline of control values is shown as a black line, and glycosaminoglycan (GAG) values of patients with mucopolysaccharidosis (MPS) are shown as plus marks.

**Figure 2 diagnostics-11-01563-f002:**
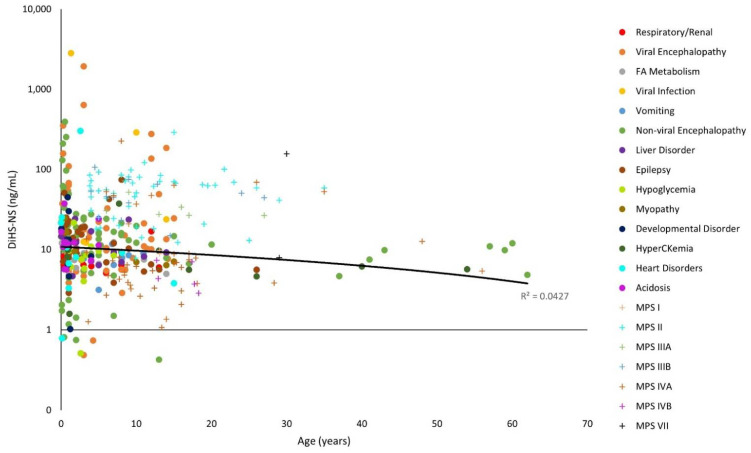
*N*-sulfated heparan sulfate (∆DiHS-NS) levels with age on a logarithmic scale. The linear trendline of control values is shown as a black line, and glycosaminoglycan (GAG) values of patients with mucopolysaccharidosis (MPS) are shown as plus marks.

**Figure 3 diagnostics-11-01563-f003:**
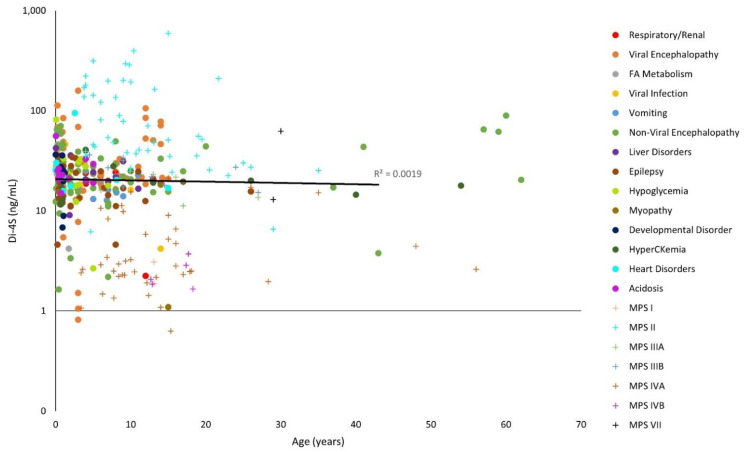
Dermatan sulfate (∆Di-4S) levels with age on a logarithmic scale. The linear trendline of control values is shown as a black line, and glycosaminoglycan (GAG) values for patients with mucopolysaccharidosis (MPS) are shown as plus marks.

**Figure 4 diagnostics-11-01563-f004:**
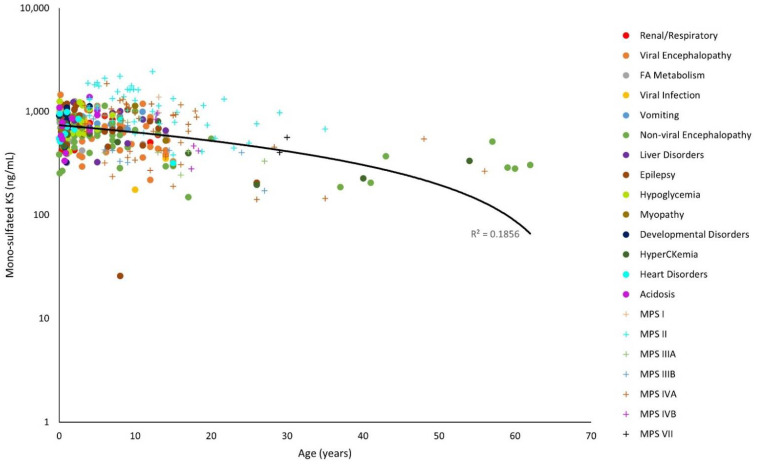
Mono-sulfated keratan sulfate (KS) levels with age on a logarithmic scale. The linear trendline of control values is shown as a black line, and glycosaminoglycan (GAG) values of patients with mucopolysaccharidosis (MPS) are shown as plus marks.

**Figure 5 diagnostics-11-01563-f005:**
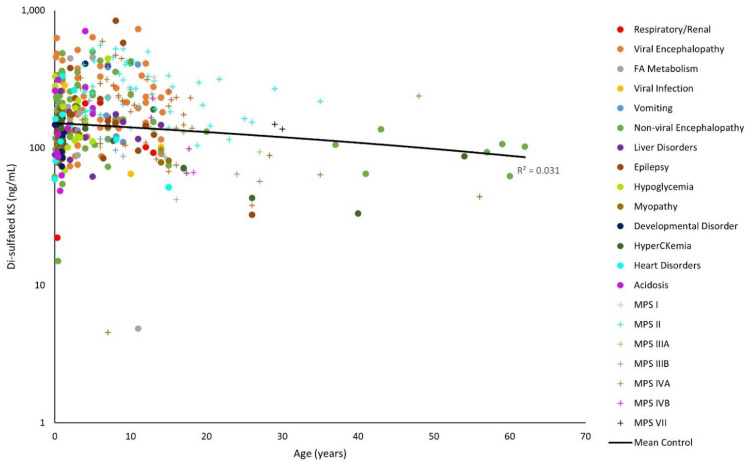
Di-sulfated keratan sulfate (KS) levels by age on a logarithmic scale. The linear trendline of control values is shown as a black line, and glycosaminoglycan (GAG) values of patients with mucopolysaccharidosis (MPS) are shown as plus marks.

**Figure 6 diagnostics-11-01563-f006:**
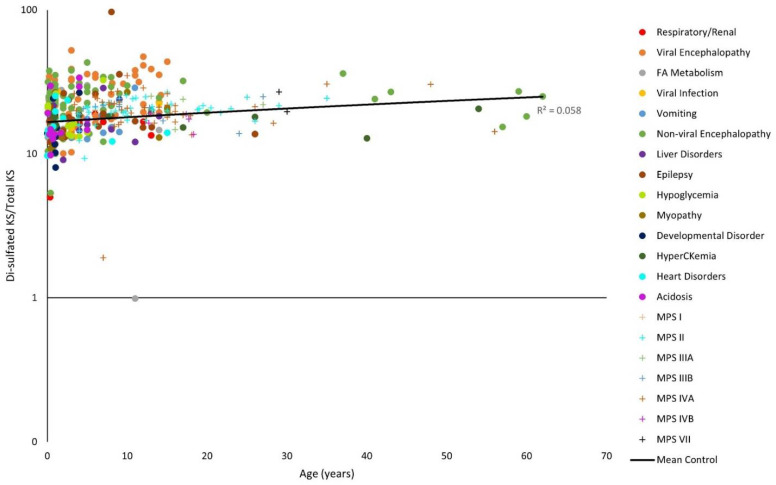
Ratio of di-sulfated keratan sulfate (KS) to total KS levels with age on a logarithmic scale. The linear trendline of control values is shown as a black line, and glycosaminoglycan (GAG) values for patients with mucopolysaccharidosis (MPS) are shown as plus marks.

**Table 1 diagnostics-11-01563-t001:** Types of mucopolysaccharidosis (MPS) and ages of patients with MPS.

Diagnosis	Total	0–2.9	3–4.9	5–9.9	10–14.9	15–19.9	20+
MPS I	2	--	--	--	2	--	--
MPS II	47	--	7	17	9	7	7
MPS IIIA	6	--	--	2	1	2	1
MPS IIIB	11	--	2	5	2	--	2
MPS IVA	42	--	3	15	8	11	5
MPS IVB	5	--	--	--	2	3	--
MPS VII	2	--	--	--	--	--	2

**Table 2 diagnostics-11-01563-t002:** Number of control values for each glycosaminoglycan (GAG) broken into age group. The GAG measurements included are *O*-sulfated heparan sulfate (DiHS-0S), *N*-sulfated heparan sulfate (DiHS-NS), dermatan sulfate (Di-4S), mono-sulfated keratan sulfate (Mono-S KS), di-sulfated keratan sulfate (Di-S KS), and the ratio of di-sulfated keratan sulfate to total keratan sulfate (Di-S KS/Total KS).

GAG	Total	0–2.9	3–4.9	5–9.9	10–14.9	15–19.9	20+
DiHS-0S	260	125	33	52	27	8	15
DiHS-NS	274	142	34	49	27	7	15
Di-4S	283	149	35	57	27	7	8
Mono-sulfated KS	313	166	36	58	32	8	13
Di-sulfated KS	282	146	32	50	31	8	15
Di-S KS/Total KS	264	139	29	47	28	7	14

**Table 3 diagnostics-11-01563-t003:** Median along with minimum and maximum glycosaminoglycans (GAGs) for control values and patients with viral encephalopathy (VE), non-viral encephalopathy (NVE), or mucopolysaccharidosis (MPS). The *p*-value for the non-parametric Kruskal–Wallis one-way analysis of variance (ANOVA) test run between the control, encephalopathy, and MPS groups is shown for each GAG at each age group. Statistically significant *p*-values for the Kruskal–Wallis ANOVA test are marked with an asterisk. Statistical differences between the control and encephalopathy groups or the control and MPS groups were determined with Dunn post-hoc testing and are marked with §. A significance level of 0.05 was used for both tests.

		DiHS-0S	DiHS-NS	Di-4S	Di-Sulfated KS	Mono-Sulfated KS	Di-Sulfated/ Total KS
Age	Diagnosis	Median (Min, Max)	Median (Min, Max)	Median (Min, Max)	Median (Min, Max)	Median (Min, Max)	Median (Min, Max)
**0**–**2.9**	Control	74.5 (21, 137)	11 (1, 28)	21.4 (2, 40)	129.1 (15, 285)	731 (253, 1193)	14.4 (5, 27)
	NVE	116.7 (24, 1129) §	17.6 (1, 393) §	26.5 (2, 70) §	184.1 (15, 488) §	633.5 (253, 1201)	21.9 (5, 38) §
	VE	236.1 (40, 928) §	37.5 (4, 353) §	38.1 (5, 113) §	305.3 (74, 630) §	825.2 (375, 1458)	23.9 (10, 35) §
	MPS I	--	--	--	--	--	--
	MPS II	--	--	--	--	--	--
	MPS IIIA	--	--	--	--	--	--
	MPS IIIB	--	--	--	--	--	--
	MPS IVA	--	--	--	--	--	--
	MPS IVB	--	--	--	--	--	--
	MPS VII	--	--	--	--	--	--
	*p* value	<0.001 *	0.003 *	<0.001 *	<0.001 *	0.06	<0.001 *
**3**–**4.9**	Control	62.9 (3, 99)	7.3 (0, 19)	20.1 (1, 33)	143.7 (75, 288)	687.9 (293, 1199)	16.2 (10, 29)
	NVE	64.7 (58, 189)	8.9 (5, 28)	25.1 (15, 41)	278.5 (99, 355)	622.4 (400, 853)	25 (20, 38) §
	VE	64.9 (3, 3227)	8.9 (0, 1934)	15.9 (1, 158)	179.2 (75, 516)	659.4 (293, 1042)	18.1 (10, 52)
	MPS I	--	--	--	--	--	--
	MPS II	140.5 (65, 222) §	55 (18, 85) §	137.3 (6, 223) §	187.3 (172, 403)	1306.7 (915, 1879) §	15.8 (9, 19)
	MPS IIIA	--	--	--	--	--	--
	MPS IIIB	258.2 (189, 327) §	79.7 (53, 106) §	30.3 (24, 36)	191.5 (162, 221)	775.6 (702, 850)	19.7 (19, 21)
	MPS IVA	38.2 (16, 65)	9.2 (1, 14)	2.4 (1, 3)	272 (160, 323)	761 (643, 1142)	22 (20, 26)
	MPS IVB	--	--	--	--	--	--
	MPS VII	--	--	--	--	--	--
	*p* value	0.001 *	0.001 *	0.002 *	0.157	0.002 *	0.005 *
**5**–**9.9**	Control	63.7 (20, 97)	7.9 (1, 15)	19.8 (2, 33)	160.4 (62, 330)	694.6 (284, 1068)	18 (12, 31)
	NVE	78 (32, 140)	14.3 (1, 24) §	21 (2, 50)	173.7 (73, 501)	613.4 (284, 1135)	26.6 (12, 43) §
	VE	67.2 (30, 123)	8.8 (3, 33)	22.5 (16, 33)	160.4 (87, 639)	669.9 (405, 1140)	22.3 (16, 36) §
	MPS I	--	--	--	--	--	--
	MPS II	147.9 (35, 393) §	50.4 (8, 99) §	89.7 (25, 315) §	402.3 (174, 557) §	1556.9 (795, 2198) §	20.9 (16, 26)
	MPS IIIA	141.8 (93, 190)	41.3 (30, 52)	24.1 (16, 32)	158.6 (102, 216)	594.7 (401, 788)	20.8 (20, 22)
	MPS IIIB	131.2 (83, 214) §	45.1 (23, 75) §	19.2 (16, 34)	114.8 (87, 207)	418.3 (322, 730)	21.5 (21, 23)
	MPS IVA	35.3 (11, 449)	5.2 (3, 226)	2.9 (1, 25) §	241.2 (5, 595) §	997 (235, 1860)	22.6 (2, 27)
	MPS IVB	--	--	--	--	--	--
	MPS VII	--	--	--	--	--	--
	*p* value	<0.001 *	<0.001 *	<0.001 *	<0.001 *	<0.001 *	0.015 *
**10**–**14.9**	Control	67.2 (8, 97)	8.2 (4, 13)	20.1 (1, 28)	124.8 (5, 283)	525.8 (218, 840)	18.7 (1, 39)
	NVE	88.2 (13, 159)	10 (0, 20)	19.9 (15, 33)	199 (96, 427)	638.7 (296, 1006)	25.7 (20, 30)
	VE	96.5 (62, 827) §	13.5 (8, 278) §	48.7 (18, 106) §	223.2 (91, 731) §	447.6 (218, 1189)	33.4 (19, 47) §
	MPS I	55.5 (48, 63)	9.4 (9, 10)	2.5 (2, 3)	277.1 (227, 327)	1176.6 (973, 1380)	19 (19, 19)
	MPS II	73.1 (41, 301)	61.6 (14, 122) §	48 (27, 398) §	270.3 (139, 501) §	1000.6 (613, 2442) §	20.8 (17, 25)
	MPS IIIA	94.3 (94, 94)	27.6 (28, 28)	23 (23, 23)	135.1 (135, 135)	401.4 (401, 401)	25.2 (25, 25)
	MPS IIIB	134.5 (56, 213)	40 (15, 65)	19.9 (18, 22)	166.2 (86, 246)	675.1 (421, 929)	19 (17, 21)
	MPS IVA	26.3 (13, 130)	4.4 (1, 47)	2.3 (1, 15) §	163.2 (83, 313)	534 (270, 1181)	20.9 (16, 35)
	MPS IVB	26.2 (22, 30)	5.8 (4, 7)	2 (2, 2)	197.3 (165, 229)	903.9 (842, 966)	17.8 (16, 19)
	MPS VII	--	--	--	--	--	--
	*p* value	0.004 *	<0.001 *	<0.001 *	0.037 *	0.008 *	0.029 *
**15**–**19.9**	Control	41.3 (35, 84)	6.7 (4, 15)	16.9 (1, 25)	73 (52, 256)	322.8 (149, 469)	20.1 (14, 32)
	NVE	56.2 (35, 78)	10.7 (7, 15)	20.2 (16, 25)	72.5 (70, 75)	222.7 (149, 297)	26.1 (20, 32)
	VE	84.4 (84, 84)	24.5 (24, 24)	20.8 (21, 21)	255.6 (256, 256)	329.2 (329, 329)	43.7 (44, 44)
	MPS I	--	--	--	--	--	--
	MPS II	61.1 (39, 413)	64.5 (12, 290) §	50.9 (24, 594) §	204.9 (104, 335)	787.4 (380, 1336) §	20.7 (19, 27)
	MPS IIIA	108.2 (99, 117)	30.4 (27, 34)	16.5 (11, 22)	85.7 (42, 129)	326.9 (243, 411)	19.4 (15, 24)
	MPS IIIB	--	--	--	--	--	--
	MPS IVA	30.3 (17, 129)	6.9 (2, 63)	4.7 (1, 21)	174.2 (67, 259)	885.1 (189, 1161)	18.8 (14, 26)
	MPS IVB	15.7 (13, 18) §	3.7 (3, 7)	2.9 (2, 4)	66.1 (65, 98)	416.5 (280, 464)	17.5 (14, 19)
	MPS VII	--	--	--	--	--	--
	*p* value	0.001 *	0.005 *	0.001 *	0.009 *	0.013 *	0.141
**20+**	Control	45.5 (31, 88)	6.4 (5, 12)	15.2 (4, 70)	86.8 (33, 207)	280.7 (183, 424)	18.8 (13, 30)
	NVE	52.5 (33, 88)	9.8 (5, 12)	43.9 (4, 90)	103.9 (62, 136)	296 (186, 547)	24.6 (15, 36)
	VE	--	--	--	--	--	--
	MPS I	--	--	--	--	--	--
	MPS II	31.9 (27, 177)	58.9 (13, 101) §	25.6 (7, 210)	163.1 (115, 316) §	677.5 (444, 1317) §	20.9 (17, 25)
	MPS IIIA	85.1 (85, 85)	26.5 (27, 27)	13.6 (14, 14)	93.6 (94, 94)	331 (331, 331)	22.1 (22, 22)
	MPS IIIB	184.9 (140, 230)	47.5 (44, 51)	21.3 (15, 27)	60.7 (57, 64)	286.2 (171, 401)	19.4 (14, 25)
	MPS IVA	39.4 (16, 141)	12.6 (4, 69)	4.4 (2, 17)	63.7 (38, 239)	265.6 (141, 544)	21.3 (14, 31)
	MPS IVB	--	--	--	--	--	--
	MPS VII	238.6 (37, 440)	82.6 (8, 157)	37.7 (13, 62)	142.9 (137, 148)	482.2 (402, 562)	23.3 (20, 27)
	*p* value	0.302	0.003 *	0.124	0.005 *	0.008 *	0.765

## Data Availability

All relevant data are within the manuscript. Any data interpretation is available on the request.

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
