# Peer review of "Glycosaminoglycans as Biomarkers for Mucopolysaccharidoses and Other Disorders"

_diagnostics, 2021, doi:10.3390/diagnostics11091563_

Round 1
Reviewer 1 Report
This study focuses glycosaminoglycans as biomarkers for mucopolysaccharidoses and other disorders. It is an interesting research and has a scientific contribution to the related research field. However, there are some issues to be figured out before publication. My comments are as follows:
- Is the prediction also applied to other race and gender?
- Please simplify the part of Introduction, and emphasize the novelty and significance of this study in the section of Introduction.
- Please delete the outline border of Figures.
- The captions of Figure and Table should be independent. Please give some necessary details and full names for abbreviations.
- Please add the subtitles in the section of Results.
- Please suggest the prediction accuracy and deficiency of this study.
Author Response
We thank you for your critical comments. We have responded to each comment below:
- Is the prediction also applied to other race and gender?
The control patients and patients with conditions other than MPS were obtained from a Japanese population. Therefore, although we expect that these results would be seen across other races, we cannot conclude that from the current data. The MPS data included patients from multiple races. Both male and female patients were included (see Supplementary Table 1), so that we expect these results are accurate for both sexes. The following was added to the Subjects subsection of methods:
“Both male and female patients were included. All patients were Japanese, and therefore, although we expect that we would see similar results across other populations, we cannot draw similarity and difference in the different ethnic backgrounds with the current data.” (lines 130-133)
“MPS patients were from various ethnic backgrounds and included both male and female patients, so we expect that these results would hold across various populations and sexes.” (lines 172-173)
- Please simplify the part of Introduction, and emphasize the novelty and significance of this study in the section of Introduction.
We edited some portion of the introduction and added the following sentence to the beginning of the introduction to simplify and emphasize the aim of the study:
We added the following to end of the introduction section, and edited the last two sentences (lines 103-112):
“In the present study, patients with conditions including MPS and other diseases, have been grouped according to their clinical diagnosis. The clinical diagnosis groups include respiratory disorders, renal disorders, fatty acid metabolism disorders, viral infections, vomiting disorders, liver disorders, epilepsy, hypoglycemia, myopathy, developmental disorders, hyperCKemia, heart disease, acidosis, and encephalopathy. Thus, in this study, we have aimed to investigate GAG expression patterns in blood of pediatric patients with various conditions and compare their levels of circulating GAGs to the levels of patients with MPS and age-matched controls. This is a novel study involving patients with various diseases to determine if GAGs could be used as biomarkers.”
- Please delete the outline border of Figures.
We have deleted the outlines in Figures 1-6. We also deleted gridlines, moved all the x-axes to the bottom of the graphs, simplified the axis labels, and darkened all the text to make it easier to read.
- The captions of Figure and Table should be independent. Please give some necessary details and full names for abbreviations.
The table and figure captions have been adjusted:
- Table 1. Types of mucopolysaccharidosis (MPS) and ages of patients with MPS.
- Table 2. Numbers and ages of control values in each age group.
- Table 3. Average and standard deviation of glycosaminoglycans (GAGs) for control, viral encephalopathy (VE), non-viral encephalopathy (NVE), and mucopolysaccharidosis (MPS) patients. The p-value for a one-way analysis of variance (ANOVA) test run between the control, encephalopathy, and MPS groups is shown for each GAG at each age group. Statistical differences between the control and encephalopathy groups were determined with post-hoc testing and are marked with §.
- Figure O-sulfated heparan sulfate (∆DiHS-0S) levels with age on a logarithmic scale. The linear trendline of control values is shown as a black line, and GAG values of patients with mucopolysaccharidosis (MPS) are shown as plus marks.
- Figure 2. N-sulfated heparan sulfate (∆DiHS-NS) levels with age on a logarithmic scale. The linear trendline of control values is shown as a black line, and GAG values of patients with mucopolysaccharidosis (MPS) are shown as plus marks.
- Figure 3. Dermatan sulfate (∆Di4S) levels with age on a logarithmic scale. The linear trendline of control values is shown as a black line, and GAG values of patients with mucopolysaccharidosis (MPS) are shown as plus marks.
- Figure 4. Mono-sulfated keratan sulfate (KS) levels with age on a logarithmic scale. The linear trendline of control values is shown as a black line, and GAG values of patients with mucopolysaccharidosis (MPS) are shown as plus marks.
- Figure 5. Di-sulfated keratan sulfate (KS) levels with age on a logarithmic scale. The linear trendline of control values is shown as a black line, and GAG values of patients with mucopolysaccharidosis (MPS) are shown as plus marks.
- Figure 6. Ratio of di-sulfated keratan sulfate (KS) to total KS levels with age on a logarithmic scale. The linear trendline of control values is shown as a black line, and GAG values of patients with mucopolysaccharidosis (MPS) are shown as plus marks.
- Please add the subtitles in the section of Results.
We have added the following subtitles to the results section:
- 1. Conditions with Elevated GAG Levels (line 231)
- 2. Encephalopathy GAG Levels (line 245)
- 3 GAG Levels of Various Conditions (line 273)
- 4. GAG Levels of Newborns (line 337)
- Please suggest the prediction accuracy and deficiency of this study.
This is an observational study with the aim to investigate serum GAG levels in patients with MPS and encephalopathy (viral or non-viral), as well as to compare that with the levels in control subjects. Data were collected retrospectively, and mean GAG levels were compared between 10 groups of MPS (I – VII), encephalopathy (viral or non-viral), and control subjects by age groups (0-2.9, 3-4.9, 5-9.9, 10-14.9, 15-19.9, 20+). As a retrospective study, the design is imbalanced and there is limited or no information in several groups across age groups. The study used One-Way analysis of variance (ANOVA) to compare mean GAG levels between groups. The highly significant p-values of various One-Way ANOVA showed the remarkable differences in mean levels between groups. The One-Way ANOVA uses the F-ratio to test the overall fit of a linear model and the estimated means are the predicted values. The highly statistical significance of ANOVA test indicates that variations in GAG levels are mostly explained by the variation due to groups rather than residuals. Therefore, the findings of our study are important to discover the differences in GAG levels in various study groups involving rare diseases of this study, although it may carry various well-known limitations of the observational studies. Despite limitations, data from observational studies have become an increasingly important source of evidence (Lighthelm et al). To reflect this, we have added the following to the Statistical Analysis section of the Methods:
“This is an observational study with the aim to investigate serum GAG levels in patients with MPS and encephalopathy (viral or non-viral), as well as to compare that with the levels in control subjects. Patients were grouped according to diagnosis or condition. Since GAG levels are also influenced by age, patients were then divided into the following age groups: x < 3 years, 3 ≤ x < 5 years, 5 ≤ x < 10 years, 10 ≤ x < 15 years, and over 15 years of age. Mean GAG levels were compared between ten groups for each GAG at each age group for the control group, viral encephalopathy patients, non-viral encephalopathy patients, MPS I patients, MPS II patients, MPS IIIA patients, MPS IIIB patients, MPS IVA patients, MPS IVB patients, and MPS VII patients.
We also used a one-way analysis of variance (ANOVA) to compare mean GAG levels between groups. The one-way ANOVA used an F-ratio to test the overall fit of a linear model and the estimated means of the predicted values. Any statistical significance of the ANOVA test indicates that variations in GAG levels are mostly explained by the variation due to groups rather than residuals. Additionally, post-hoc testing was used to determine statistical significance between the control and encephalopathy groups. A p-value of less than 0.5 was considered significant. Despite well-known limitations, data from observational studies have become an increasingly important source of evidence, and thus these results have relevance [19].” (lines 211-228)
Reviewer 2 Report
This retrospective study aims to investigate the association between serum GAG levels and various clinical conditions. It is an interesting study with clear methodology and nicely written discussion. What is also important is that the limitations are clearly presented. I have no further comments
Author Response
Response; We thank you for your positive comments.
Reviewer 3 Report
This study compares the levels of GAGs that are used as markers in MPS with those found in patients suffering from many different clinical conditions. Potentially the study could have relevance but unfortunately, for each disease studied, the number of patients is so small that statistical analysis cannot be conducted. In view of this, the data collected so far cannot, in my opinion, be published. The authors should enlarge the number of patients even for the most promising diseases (viral and non viral encephalopathy) and carry out a statistical analysis to evaluate GAGs as biomarkers.
Author Response
We thank you for your critical comments and suggestions. We have addressed your main concerns below:
- We do agree that larger sample sizes would yield more conclusive results, and we have acknowledged this as a limitation of the study in our discussion:
“There are several limitations for the current study, including that it was a retrospective study and most sample groups were small, affecting the statistical analysis. Further experiments with larger sample sizes are necessary to confirm these results.” (lines 460-462)
While larger sample sizes are necessary for most conditions, we did see some statistical differences in the encephalopathy group (see Table 3), which we also have shown it in the results:
“There is some variation in average GAG levels across age groups, especially in the two oldest age groups. However, there are significant differences in DiHS-0S, DiHS-NS, Di-4S, di-sulfated KS, and di sulfated KS/total KS between the control, viral encephalopathy, non-viral encephalopathy, MPS I, MPS II, MPS IIIA, MPS IIIB, MPS IVA, and MPS IVB. Of particular note is the 0-2.9 age group, which only contained the control and two encephalopathy groups. In this comparison, there is a significant difference in DiHS-0S, DiHS-NS, Di-4S, di-sulfated KS, and di sulfated KS/total KS between the control group, viral encephalopathy group, and non-viral encephalopathy group [Table 3]. Using post-hoc testing for pairwise comparisons to the control group, viral encephalopathy had significantly higher DiHS-0S and DiHS-NS means for the 0-2.9 age group and the 10-14.9 age group, significantly higher Di-4S and for the 0-2.9 age group, significantly higher di-sulfated KS for the 0-2.9 age group, and a significantly higher ratio of di-sulfated KS to total KS for the 0-2.9 age group, the 5-9.9 age group, the 10-14.9 age group, and the 15-19.9 age group. Non-viral encephalopathy had significantly higher DiHS-0S and Di-HS-NS for the 0-2.9 age group, significantly higher Di-4S for the 0-2.9 age group, significantly higher di-sulfated KS for the 0-2.9 age group, and a significantly higher ratio of di-sulfated KS to total KS for the 0-2.9 age group and the 5-9.9 age group. None of the age groups had a statistically significant difference in average mono-sulfated KS values for either type of encephalopathy.” (lines 263-281)
While additional experiments with increased sample sizes will affirm the current findings, we believe that the preliminary findings in this experiment do have novelty since we have found statistical differences between the control group and both encephalopathy groups in some ages.
- In response to the suggestion that the introduction could be improved, we clarified the aim of the study and emphasized the novelty and significance of our study in the last paragraph of the introduction:
“In the present study, patients with conditions including MPS and other diseases, have been grouped according to their clinical diagnosis. The clinical diagnosis groups include respiratory disorders, renal disorders, fatty acid metabolism disorders, viral infections, vomiting disorders, liver disorders, epilepsy, hypoglycemia, myopathy, developmental disorders, hyperCKemia, heart disease, acidosis, and encephalopathy. Thus, in this study, we have aimed to investigate GAG expression patterns in blood of pediatric patients with various conditions and compare their levels of circulating GAGs to the levels of patients with MPS and age-matched controls. Thus, this is a novel study involving patients with various diseases to determine if GAGs could be used as biomarkers.” (lines 104-113)
- In response to the clarity of results needing to be improved, we have done the following:
- We adjusted the figures by removing the borders around them, deleting the gridlines, moving all the x-axes to the bottom of the graphs, simplifying the axis labels, and darkening all the text to make it easier to read.
- Added subtitles to the results section.
- Adding abbreviations and clarifying details to the table and figure captions.
- Lastly, in response to the conclusions needing to be improved to support the results, we have edited both the discussion and conclusion section and added the following paragraph to the conclusion:
“Patients with MPS are known to have high levels of serum GAGs due to accumulation of GAGs in lysosomes. We have identified that some patients with other conditions have the elevation of GAGs. Especially, viral and non-viral encephalopathies are associated with elevated GAG levels. The mechanism of GAG elevation in patients with encephalopathy remains unknown. This study is retrospective and limited by the sample sizes of each disease group. Therefore, the results give a preliminary look into various conditions with an association with GAG elevation. Further studies into the association with disease severity, stage, and therapeutic effect are required before GAGs can be established as biomarkers for the diagnosis and prognosis of each disease. These findings create a foundation for further exploration of elevated GAGs in diseases beyond MPS.” (lines 490-507)
Reviewer 4 Report
The authors present the results of a retrospective study aiming to identify the role of glycosaminoglycans as biomarkers for several metabolic disorders.
A. Major comments:
1. My major concern is that in most cases the comparisons were between completely unbalanced groups. This could completely invalidate the results.
2. You applied ANOVA to compare many groups. This is true, but have you also checked the assumptions of ANOVA to ensure the robustness of the results?
First, you need to check that the assumptions of ANOVA (normality of data, similar variances between compared groups) are met, and then apply them. If you find that the above assumptions are met, you can apply parametric methods (such as ANOVA). However, if the assumptions are not met, the results obtained cannot be considered reliable.
3. Figures 1-6 are plotted on a semi-logarithmic scale, and a pictorial representation of a linear trend can be seen in this arrangement.
(a) Please justify why the Y-axis (concentration of markers) has been log-transformed.
b) Give the correlation coefficient to show the degree of linearity (also enter this value in the graph).
c) Remember that any linear relationship (positive or negative) found in the semi-log graph implies an exponential relationship between the concentration of the marker and age.
B. Minor comments:
1. Lines 220-222: please move to the Discussion section
2. For post-hoc analyses, please clarify which post-hoc method you used
3. Table 2: define abbreviations
4. Table 3: the asterisk (*) next to some values is not explained what it stands for
Author Response
- Major comments:
- My major concern is that in most cases the comparisons were between completely unbalanced groups. This could completely invalidate the results.
Response: We thank this reviewer for raising the issue of unbalanced data. Although almost always unbalanced, naturally collected data such as electronic health records have shown great utility in learning healthcare systems and precision medical decisions. The naturally collected data of this study would be able to provide valuable information about a very rare disease. We agree with this reviewer about challenges of analyzing these data and we used non-parametric methods of analyses with a hope to handle limitations related to unbalanced data. Also, please not that conclusions of our results in parametric analyses didn’t change much for using non-parametric methods. This also indicates that the analyses of data have produced reasonably valid results.
- You applied ANOVA to compare many groups. This is true, but have you also checked the assumptions of ANOVA to ensure the robustness of the results?
First, you need to check that the assumptions of ANOVA (normality of data, similar variances between compared groups) are met, and then apply them. If you find that the above assumptions are met, you can apply parametric methods (such as ANOVA). However, if the assumptions are not met, the results obtained cannot be considered reliable.
Response: We thank this reviewer again. We checked and found that all assumptions did not meet for every analysis. For this reason, we performed the alternative non-parametric Kruskal-Wallis ANOVA for comparing GAG between groups and presented the results along with median (min, max) in Table 3.
- Figures 1-6 are plotted on a semi-logarithmic scale, and a pictorial representation of a linear trend can be seen in this arrangement.
(a) Please justify why the Y-axis (concentration of markers) has been log-transformed.
Response: This was because of extreme outliers, making the distribution very difficult to see without the log-transformation. - b) Give the correlation coefficient to show the degree of linearity (also enter this value in the graph).
Response: The coefficients of determination have been added to the graphs. Note that they are all low, but we were not searching for an exponential relationship (because GAGs are not expected to increase exponentially with age), rather displaying the average GAG level through various ages.
c) Remember that any linear relationship (positive or negative) found in the semi-log graph implies an exponential relationship between the concentration of the marker and age.
Response: see above.
Minor comments:
1. Lines 220-222: please move to the Discussion section
Response: These were moved to lines 340-342 in the discussion.
- For post-hoc analyses, please clarify which post-hoc method you used
Response: We performed non-parametric Mann-Whitney U test to compare the median between groups of pairs. We adjusted the level of significance by dividing the number of comparisons for each analysis. In particularly, we were interested in comparing control vs NVE or control vs VE. We used a level of significance of 0.025 for these two comparisons.
- Table 2: define abbreviations
Response: The caption for Table 2 has been adjusted accordingly:
“Table 2. Number of control values for each glycosaminoglycan (GAG) broken into age group. The GAG measurements included are O-sulfated heparan sulfate (DiHS-0S), N-sulfated heparan sulfate (DiHS-NS), dermatan sulfate (Di-4S), mono-sulfated keratan sulfate (Mono-S KS), di-sulfated keratan sulfate (Di-S KS), and the ratio of di-sulfated keratan sulfate to total keratan sulfate (Di-S KS/Total KS).”
- Table 3: the asterisk (*) next to some values is not explained what it stands for
Response: The caption for table 3 has been adjusted:
“Table 3. Average and standard deviation of glycosaminoglycans (GAGs) for control, viral encephalopathy (VE), non-viral encephalopathy (NVE), and mucopolysaccharidosis (MPS) patients. The p-value for a one-way analysis of variance (ANOVA) test run between the control, encephalopathy, and MPS groups is shown for each GAG at each age group. Statistically significant p-values for the ANOVA test are marked with an asterisk. Statistical differences between the control and encephalopathy groups were determined with post-hoc testing and are marked with §.”
Round 2
Reviewer 1 Report
It is clear that authors have done a substantive revision. I am satisfactory about the revision.
Author Response
We thank you for your critical comments and suggestions. We have addressed your concerns below:
We do agree that larger sample sizes would yield more conclusive results, and we have acknowledged this as a limitation of the study in our discussion. However, we disagree with the reviewer’s comment in some portion. Our study includes the largest sample number compared to the previous studies. Therefore, 51 patients were diagnosed with viral encephalopathy, and 69 patients were diagnosed with non-viral encephalopathy. This was enough number for the first paper to show the statistical difference. We have clarified it as follows.
“There are several limitations for the current study. First, it was a retrospective study. Although encephalopathy group included the largest analysis of the blood sample number (n = 120 patients; viral, 51, non-viral, 69) compared to the previous studies in other disorders (n = 8-47 patients) [8-12,25,28,30,31], most sample groups were small, affecting the statistical analysis.”
As long as data support statistical analysis, we are allowed to a statistical comparison (see Table 3). While we compare the location (e.g., mean) between two groups, we account for the variability to achieve the significance of the comparison. In the planned study, the sample size estimation is performed mainly to detect the difference in mean or other estimates (between groups or aspects under investigation) significantly. In this study, many comparisons achieve high significance indicating a substantial difference between groups in the response variables. However, considering an observational study with a limited size of unbalanced groups, we need to be careful in interpreting results and the generalization of findings of the study. The results in the manuscript are interpreted with extreme caution. Therefore, there should not have any statistical concern.
Reviewer 4 Report
Thank you for the replies to my remarks.
However, these responses are not sufficient and further clarification is needed.
Comments on the answers given:
1. The answer is quite general. If the data is unbalanced, there are appropriate statistical methods that can be used. Even if you have used non-parametric methods, you need to ensure that the distribution of variables (shape) is similar beforehand.
2. OK
3. Please include this clarification in the text (both in the Results section and in the Discussion).
Minor comments:
1. OK
2. Post-hoc tests are non-parametric, but the reverse is not always true.
The MW is a non-parametric text, but it is not one of the post-hoc tests and is not appropriate to be used in post-hoc analyses.
I recommend that you use the results of the Tukey or Dunn test.
Please indicate which software (and version) was used for the statistical analyses.
Author Response
Thank you for the replies to my remarks.
However, these responses are not sufficient, and further clarification is needed.
We thank the reviewer for their thorough comments and suggestions. We have substantially adjusted the statistic methods and detailed them in the test of the manuscript.
Comments on the answers given:
1. The answer is quite general. If the data is unbalanced, there are appropriate statistical methods that can be used. Even if you have used non-parametric methods, you need to ensure that the distribution of variables (shape) is similar beforehand.
We thank this reviewer for their comment and agree that the distribution of variables should be similar. An examination of box plots exhibited a similar shape in the distribution of GAG, except a few outliers, across diagnostic groups for each age category (see below as an example at age 0-2.9; PDF file). A non-parametric Kruskal-Wallis one-way analysis of variance (ANOVA) was performed to handle outliers (a few) of this unbalanced dataset.
We added this clarification in the Methods section:
“Examination of box plots exhibited a similar shape in the distribution of GAGs, except a few sparse outliers, across diagnostic groups for each age category. A non-parametric Kruskal-Wallis one-way analysis of variance (ANOVA) was performed to compare median GAG levels between ten groups for each GAG at each age group for the control group, viral encephalopathy patients, non-viral encephalopathy patients, MPS I patients, MPS II patients, MPS IIIA patients, MPS IIIB patients, MPS IVA patients, MPS IVB patients, and MPS VII patients.” (lines 211-217)
- OK
- Please include this clarification in the text (both in the Results section and in the Discussion).
The clarification has been added:
“The y-axis for figures 1-6 has been log-transformed due to extreme outliers. Without the transformation, outliers made it difficult to see the distribution of GAG measurements.” (lines 268-270)
“Note that the control GAG values were represented with a linear model [Figures 1-6]. The y-axes were log-transformed, so any linear relationship on the graphs would have implied an exponential relationship. The coefficients of determination were low for all models, but this is not an issue, as we were not searching for an exponential relationship since GAGs do not increase exponentially with age; rather, displaying the average GAG level across ages.” (lines 343-348)
Minor comments:
1. OK
- Post-hoc tests are non-parametric, but the reverse is not always true.
The MW is a non-parametric text, but it is not one of the post-hoc tests and is not appropriate to be used in post-hoc analyses.
I recommend that you use the results of the Tukey or Dunn test.
Please indicate which software (and version) was used for the statistical analyses.
Thanks again to this reviewer for this recommendation. We changed the post hoc testing to a Dunn test for multiple comparisons. The statistical software packages R, version 3.5.2, and SPSS, version 27, were used for data analyses.
Table 3, including its caption, was edited accordingly. The text accompanying table 3 was adjusted as well (lines 251-266).

Round 3
Reviewer 4 Report
The answers are satisfactory. I have no further comments to make.